# Detection and Classification of Unannounced Physical Activities and Acute Psychological Stress Events for Interventions in Diabetes Treatment

Mohammad Reza Askari [1], Mahmoud Abdel-Latif [1], Mudassir Rashid [1], Mert Sevil [1] and Ali Cinar [1,2,*]

1 Department of Chemical and Biological Engineering, Illinois Institute of Technology, Chicago, IL 60616, USA
2 Department of Biomedical Engineering, Illinois Institute of Technology, Chicago, IL 60616, USA
* Correspondence: cinar@iit.edu

**Abstract:** Detection and classification of acute psychological stress (APS) and physical activity (PA) in daily lives of people with chronic diseases can provide precision medicine for the treatment of chronic conditions such as diabetes. This study investigates the classification of different types of APS and PA, along with their concurrent occurrences, using the same subset of feature maps via physiological variables measured by a wristband device. Random convolutional kernel transformation is used to extract a large number of feature maps from the biosignals measured by a wristband device (blood volume pulse, galvanic skin response, skin temperature, and 3D accelerometer signals). Three different feature selection techniques (principal component analysis, partial least squares–discriminant analysis (PLS-DA), and sequential forward selection) as well as four approaches for addressing imbalanced sizes of classes (upsampling, downsampling, adaptive synthetic sampling (ADASYN), and weighted training) are evaluated for maximizing detection and classification accuracy. A long short-term memory recurrent neural network model is trained to estimate PA (sedentary state, treadmill run, stationary bike) and APS (non-stress, emotional anxiety stress, mental stress) from wristband signals. The balanced accuracy scores for various combinations of data balancing and feature selection techniques range between 96.82% and 99.99%. The combination of PLS–DA for feature selection and ADASYN for data balancing provide the best overall performance. The detection and classification of APS and PA types along with their concurrent occurrences can provide precision medicine approaches for the treatment of diabetes.

**Keywords:** recurrent neural network; long short-term memory; feature selection; imbalanced data; activity recognition; acute psychological stress detection; precision medicine; diabetes

## 1. Introduction

Meals, physical activities (PA), and acute psychological stress (APS) strongly affect the blood glucose concentrations (BGC) of people with diabetes. Diabetes is a worldwide public health problem, with an estimated 537 million people living with diabetes in 2021 [1]. While people with diabetes can take appropriate actions to prevent large variations in BGCs that can lead to hypoglycemia and hyperglycemia (glucose levels lower or higher than the safe zone, respectively), they may not be able to make adjustments when PA and APS are unexpected or unplanned. People with Type 1 diabetes (T1D) regulate their BGC by manipulating insulin injections or infusions (by an insulin pump) to subcutaneous tissue. There is a significant time delay before insulin in the subcutaneous tissue diffuses and reaches the bloodstream to affect BGC (approximately 45 min or more). Consequently, a significant amount of time is lost in making adjustments if the decision is solely based on glucose concentration readings, for example, from a continuous glucose monitoring (CGM) system. The delay in recognizing the presence of these factors (PA and APS) that disrupt glucose homeostasis can be reduced by interpreting physiological signals measured by wearable devices commonly used in free living, such as wristbands. The information

captured from wearable devices can be used for better manual adjustment of insulin dosing or incorporated into the control algorithms of artificial pancreas systems in order to improve the regulation and time-in-range of BGC [2,3].

Decisions about the presence of PA or APS are difficult to make based solely on direct signals from wearable devices. For example, both blood volume pulse (BVP) and heart rate (HR) increase with PA and APS. Erroneous selection of the cause of BGC variations can have severe consequences, as most PA reduces BGC while APS increases it, with the former leading to hypoglycemia and the latter to hyperglycemia. In the former case, intervention is intended to reduce insulin flow, while in the later it is intended to increase infusion. Erroneous determination of the source cause (PA or APS) and the wrong intervention can accentuate undesirable changes in BGC variation.

Several devices and algorithms have been reported for characterizing APS and PA in clinical environments with high accuracy [4–13]. In free-living environments, sensors are limited to those that can be worn comfortably and accepted as part of daily wear. Recent advances in wearable devices have provided wristbands with three-axis accelerometers (ACC) as well as sensors for measuring galvanic skin response (electrodermal activity) (GSR), skin temperature (ST), and blood volume pulse (to infer HR and HR variability). The data reported by these sensors must be preprocessed in order to reduce sensor noise, handle missing values, and accommodate imbalances in the number of data points for various types of events. For example, the amount of data during sedentary states can be much greater than during PA or during non-stress states compared to APS episodes, which can bias the models that interpret the data. As such, features must be extracted and refined in order to accurately interpret, classify, and incorporate the effects of APS and PA into treatment decisions. The detection and discrimination of APS and PA can be complemented by determining the characteristics of these psychological and physiological stressors in order to propose individualized intervention strategies and mitigate their effects by manual adjustment or automated insulin delivery by adaptive multivariable artificial pancreas systems [14–16].

We have previously reported data preprocessing, feature definition, selection of most informative features, and use of computationally efficient algorithms in multivariable statistical and machine learning (ML) techniques [17–20]. In [19], the authors developed a classification system able to distinguish five types of physical activity, such as rest, daily activities, living, running, biking, and resistance training, along with models to estimate the energy expenditure (EE) associated with diabetes therapy. Deep neural networks using the long-short-term-memory (LSTM) architecture had 94.8% classification accuracy, while various other ML techniques had varying success rates in classifying the five different PAs. In [17], PA (sedentary, treadmill running, and stationary bike) and APS (non-stress state, mental stress, and emotional anxiety) were simultaneously detected and discriminated using various classical ML algorithms. Accuracy scores over 99% for detecting PA and 92% for discriminating APS were achieved for the concurrent presence of PA and APS. Clinical data collected using the Empatica E4 wristband [21] (ACC, BVP, HR, GSR, ST) were used to assess the performance of various ML methods (Figure 1). PA categories included sedentary state, treadmill running, and stationary bicycling, while APS events included non-stress states, mental stress, and emotional anxiety.

Motivated by [17,19], this paper focuses on alternative techniques for the selection of the most informative set of features, balancing the data for various groups of activities with different data lengths, and the use of deep neural networks (NN) for detection and classification of PA and APS. The performance of principal components analysis (PCA), partial least squares–discriminant analysis (PLS-DA), and sequential forward selection (SFS) in identifying the most informative features is investigated. Then, the effects of four different approaches used for handling different data lengths of various classes by up- and down-sampling (the Adaptive Synthetic Sampling (ADASYN), random oversampling and downsampling, and cost-sensitive/weighted balancing techniques) are compared. Lastly,

we explored the performance of a neural network model using the LSTM architecture for each combination of feature selection and data balancing technique.

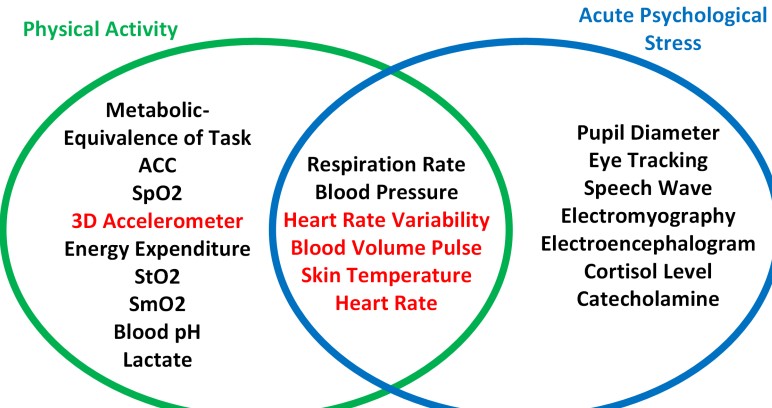

**Figure 1.** Physological variables measured by Empatica E4 for detecting PA and APS and discriminating between them (in red fonts).

The remainder of the paper is structured as follows. In the Materials and Methods section, the data sources used in this work are described along with the techniques for generating features from these data, selecting the most informative features, and balancing the effects of the different data lengths for various groups of activities, and the deep neural networks (CNN-LSTM) used for detecting and classifying various types of PA and APS events are presented. The Results section reports the performance of the algorithms developed when PA or APS occur individually or concurrently. The Discussion section presents the impact of the proposed techniques on the treatment of chronic disease and assesses the performance of various alternative techniques tested to carry out the sequence of activities involving signal processing, data rearrangement, PA and APS detection, and classification of the results. The Conclusions section summarizes the results and the potential benefits of the proposed approach for the treatment of diabetes.

## 2. Materials and Methods

The levels of several hormones, such as lactate and cortisol, can provide valuable information about the presence, duration, or intensity of PA and APS (Figure 1). Unfortunately, there are no sensors currently available to measure these hormonal changes noninvasively or minimally invasively in daily life; thus, data on physiological variables that can be noninvasively measured by a single wearable wristband device are used as surrogate information [4–13] (indicated in red in Figure 1). In this study, we used the Empatica E4 wristband, which uses an internal algorithm to measure a number of physiological variables, including ACC readings, BVP, GSR, ST, and HR, on the basis of BVP [21,22]. The E4 can store the data, which can then be downloaded as a batch. The measured physiological variables can then be used to generate informative features and train ML algorithms after preprocessing and balancing the data. The specifications of the Empatica E4 and the physiological variables it records, along with those of the Bioplux system (which provides higher accuracy PPG and ECG signals) and COSMED K5 (which provides energy expenditure information as MET values) used for comparison in this study, are summarized in Table 1.

**Table 1.** Characteristics of the measurement devices used in this study.

| Device | Sensor | Frequency of Measurement |
|---|---|---|
| Smartwatch (Empatica E4) | Gyroscope | Continuous X-axis acceleration within ±2 g with frequency of 32 Hz<br>Continuous Y-axis acceleration within ±2 g with frequency of 32 Hz<br>Continuous Z-axis acceleration within ±2 g with frequency of 32 Hz |
| | PPG | Continuous BVP signal with sampling rate of 64 Hz |
| | Infrared Thermopile | Continuous SK with the frequency of 4 Hz |
| | Electrodermal activity sensor | Continuous GSR with the frequency of 4 Hz |
| | - | Inter beat interval (IBI) calculated from BVP signal<br>(only available in the offline mode) |
| | - | Heart rate (HR) values with the frequency of 1 Hz |
| Bioplux | PPG | BVP signal with the sampling rate of 1000 Hz |
| | ECG | ECG signal with the sampling rate of 1000 Hz |
| COSMED K5 | VO2 measurement | B-B measurement of MET values each 2–4 s |

*2.1. Description of Data*

Data were collected from 34 subjects in 166 experiments while they were in a sedentary state or performing PA such as running on a treadmill or using a stationary bike (Table 2). Six were healthy subjects and 28 were people with T1D. During these experiments, three APS states were implemented: no psychological stressors (labeled as non-stress), and inducement of mental stress or emotional anxiety. The experiments were approved by the Institutional Review Board of the university. In the non-stress state, participants watched neutral videos, listened to music, read books, or surfed the internet. Emotional anxiety stress inducement included watching videos of surgeries or car crashes, meeting with supervisors, driving a car, and solving test problems within an allotted time frame. Mental stress inducement included performing the Stroop test, IQ test, mental arithmetic (mental multiplication of two-digit numbers) or a mathematics exam, and puzzle games. These APS inducements are reported in the literature to reliably induce APS when implemented in a variety of studies [8–13,23–31]. Energy expenditure was measured by a portable indirect calorimetry system (COSMED K5, Rome, Italy) [32] and compared across the non-stress, emotional anxiety, and mental stress experiments to ensure the PA was consistent across the experiments. The subjects performed similar intensity PA during treadmill run and stationary bike exercises. The Empatica E4 data were recorded in all experiments. The participants' demographic information and experiment information (APS type, PA type, duration, starting time, energy expenditure) were recorded. To assess the anxiety response of participants, state–trait anxiety inventory (STAI) [33–35] self-reported questionnaire data were collected before and after each non-stress and emotional anxiety stress inducement experiment. The STAI-T (trait version) inventory captures the anxiety or discomfort that a subject experiences on a day-to-day basis. STAI-S scores indicate temporal fear, nervousness, discomfort, and arousal of the autonomic nervous system induced by different situations that are perceived as stressful conditions [33–35]. Not all subjects performed all components of the experiments, and certain cases, such as PA without APS (non-stress), had longer durations. In addition, emotional anxiety stress and mental stress during sedentary states were much longer than during treadmill run and stationary bike PA. This necessitated balancing data sizes, as discussed later in this article.

**Table 2.** The participants' general demographic information and information on experiments conducted for data collection during sedentary state, treadmill running, and stationary bicycling.

| Summary of Participant Demographics | | | |
|---|---|---|---|
| Demographic Variable | Averaged | Min-Max | Variance |
| Age | 25.0 | 20–31 | 11.7 |
| Height (cm) | 171.2 | 154–184 | 97.1 |
| Weight (kg) | 61.9 | 49–82.9 | 123.7 |
| BMI (kg/m$^2$) | 21.1 | 16.5 | 8.2 |
| Max HR (bpm) | 195.0 | 189.0–200.0 | 11.7 |
| **PA Experiments without APS** | | | |
| PA | Number of Experiments | Number of Subjects | Minutes |
| Sedentary State | 89 | 10 | 3172 |
| Treadmill Run | 57 | 20 | 2164 |
| Stationary Bike | 61 | 19 | 1713 |
| **Sedentary State Experiments with APS Inducement** | | | |
| APS | Number of Experiments | Number of Subjects | Minutes |
| Non-Stress | 28 | 6 | 846 |
| Emotional Anxiety Stress | 29 | 9 | 1129 |
| Mental Stress | 32 | 6 | 1197 |
| **Treadmill Run Experiments with APS Inducement** | | | |
| APS | Number of Experiments | Number of Subjects | Minutes |
| Non-Stress | 28 | 20 | 1162 |
| Emotional Anxiety Stress | 12 | 12 | 676 |
| Mental Stress | 17 | 8 | 326 |
| **Stationary Bike Experiments with APS Inducement** | | | |
| APS | Number of Experiments | Number of Subjects | Minutes |
| Non-Stress | 29 | 19 | 891 |
| Emotional Anxiety Stress | 24 | 12 | 585 |
| Mental Stress | 8 | 7 | 237 |

*2.2. Data Preprocessing*

The objective of this study was to predict the occurrence of physical activity and psychological stress individually or concurrently in different subjects. All data were collected in a clinical setting. Data quality may be compromised due to factors such as sensor detachment, loss of communication, outliers, and missing information. Therefore, preprocessing of all signals was needed to improve the signal-to-noise ratio as well as to keep the valuable information from being filtered out.

The processed variables from the Empatica E4, namely, time between individual heartbeats (IBI) and HR values estimated by the E4 during physical activity and intense body movement, are contaminated by noise and motion artifacts because the photoplethysmography sensor is noise-sensitive. Hence, these variables were not taken into account when developing NN models. Instead, we used the BVP signal and processed it to generate time- and frequency-domain features that were then used in the training of the classification models.

2.2.1. Signal Windowing

The main purpose of segmenting signals is to find the correlation between the target variables and the features extracted from time-series signals. In this step, we considered a ten-second rolling window of wristband data with a step size of one second. The label of

each segment was determined from the label of the last second of the segment. Figure 2 demonstrates the notation used for labeling each segment of the signal. Figure 2 illustrates the process of stacking samples in chronological order for subsequent training of the LSTM NN model.

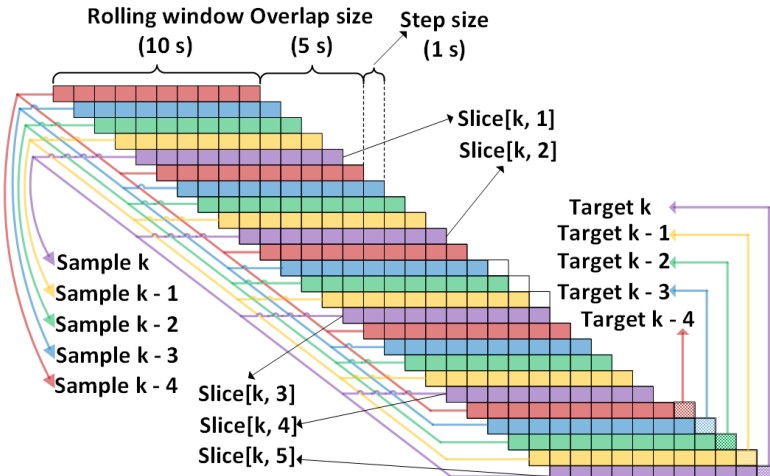

**Figure 2.** Schematic representation of windowing applied to different biosignals.

### 2.2.2. Signal Processing

Data were preprocessed differently depending on the meaningful range of variation in the biosignals. BVP is the main biosignal recorded and transmitted by the Empatica E4, providing useful information on the sympathetic and parasympathetic nervous systems that indicates episodes of autonomous and non-autonomous body behavior. Heart rate estimation, heart rate variability, and respiration rate are the most informative variables for discriminating between different PA and APS.

Depending on age, physical condition, and emotional state, the respiration rate can take values between 13 and 57 breaths per minute. Affected by these and other factors, HR values can range from 40 to 200 BPM. Therefore, HR values outside of this range are expected to be either high-frequency noise or motion artifacts. Hence, BVP signals are passed through a 4th order Butterworth bandpass filter with cutoff frequencies of 0.2–3.3 Hz in order to smooth BVP recordings and reduce noise.

A 3D accelerometer is the main signal used to detect and discriminate different types of physical activities. About 98% of the human activity frequencies lie between 0–10 Hz [36]. To filter out other frequencies that do not emanate from body movement, a simple low-pass filter or a bandpass filter with a lower frequency close to zero is be an adequate solution. We used a fourth order Butterworth bandpass filter with a cutoff frequency of 0.1–10 Hz to denoise the 3D accelerometer signals.

Electrodermal activity is typically classified into two different types, namely, the phasic skin conductance response (SCR) and the tonic skin conductance level (SCL). The SCL can be considered as the baseline of the change in electrodermal activity. It can be calculated as the rolling average of a signal over a long period of time. In the absence of external stimuli, the SCL can vary slowly over time depending on psychological condition. These changes may take as long as several minutes to occur. SCR, on the other hand, responds to rapid changes in short-term environmental stimuli, sight, noise, odor, and other factors that precede participation, such as fear, anticipation, and decision-making. The mathematical representation of this decomposition is developed by [37] as below:

$$y = r + t + \epsilon \tag{1}$$

where $y \in R^n$ denotes the original GSR signal, $t \in R^n$ is the tonic component, and $r \in R^n$ stands for the phasic behavior of the signal. The parameters of the above signal

decomposition model are identifiable by solving an optimization problem [37]. Using a linear upsampling procedure, the GSR signal was doubled to 8 Hz. The purpose of signal upsampling was to ensure that the parameters of the models fitting the signal were not obtained locally because of the short rolling windos of ten seconds. For each rolling window, the phasic and tonic components were extracted and feature variables from each were extracted. Figure 3 summarizes the signal preprocessing pipeline for each signal.

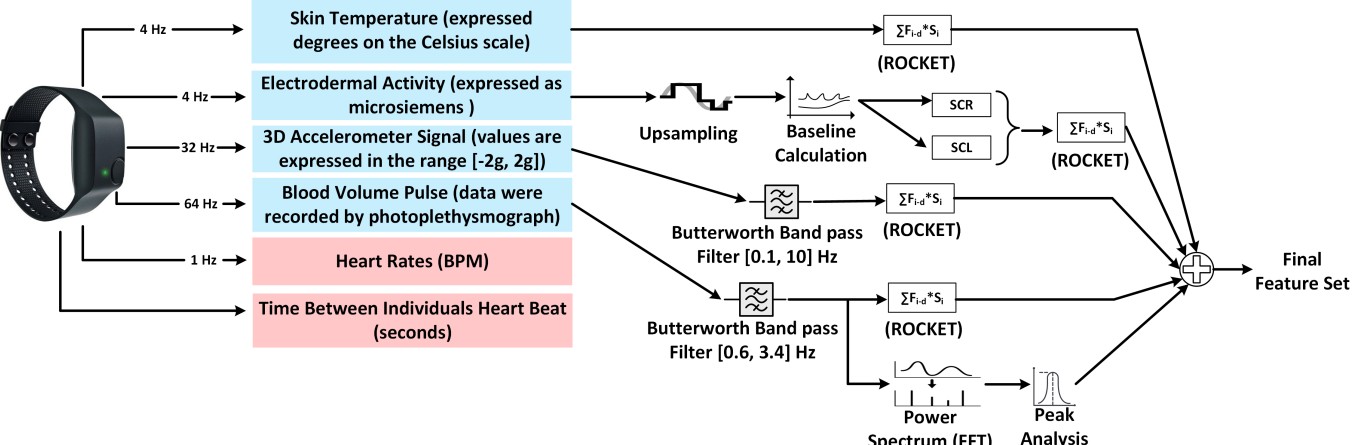

**Figure 3.** The signal processing pipeline for ROCKET feature extraction.

## 2.3. Feature Extraction

The transformation of time-series data into reconstructed feature maps relevant to the target variable is a crucial step in data preparation. This step can be achieved using deep 1D convolutional LSTM NN models [38]. The main drawback is that a large number of data samples are required, and integrating convolutional layers with LSTM architecture cannot be done efficiently with GPUs. Therefore, we employed random convolutional kernel transformation (ROCKET) [39] to extract a large number of features from time-series signals [40]. ROCKET extracts features by generating random convolutional kernels with random length, weight, bias, dilation, and padding. Although at some points using ROCKET seems similar to using convolutional and pooling layers in deep NN models, the parameters of kernel filters are not trainable. Additionally, they run faster, are resistant to dilation, and are more flexible by applying convolutional kernels with different sizes, padding, etc. Equation (2) was used for extracting dilated convolutional-based feature maps from each segment of biosignals by calculating the maximum and the proportion of positive values of the filtered signal [39,41]:

$$F(s) = (X_D \times f) = \sum_{i=0}^{m-1} f(i).x_D(s - d.i), \quad D \in \{BVP, Acc_x, Acc_y, Acc_z, SCR, SCL, ST\} \quad (2)$$

where the 1D signal $X \in \mathbb{R}^{10 \times fs_I}$ and the kernel filter $f : 0, ..., m - 1 \to \mathbb{R}$. The length $m$ of each kernel filter is selected as $2 \times fs_I$, where $fs_I$ represents the sampling rate of biosignal $I$ and the variable $d$ represents the dilation factor.

In addition to 1D kernel features, BVP signals were transformed to frequency domain by using fast Fourier transform to extract the variations of peaks orthogonal to the 3D accelerometers, applying the following equation [20,42]:

$$N_{BVP \perp Acc_t} = N_{BVP} \prod_{t \in x,y,z} \left( I_{(n_{hf} - n_{lf} + 1)} - \frac{N_{Acc_t} N_{Acc_t}^T}{N_{Acc_t}^T N_{Acc_t}} \right) \quad (3)$$

where $N_{BVP}$ and $N_{Acc_t}, t \in x, y, z$ denote the normalized power spectrum of the BVP and 3D accelerometer signals, respectively, $I_{(n_{hf} - n_{lf} + 1)}$ is the identity matrix, and $n_{hf}, n_{lf} > 0$ are indexes of spectral bins expressed in BPM and corresponding to the highest and lowest

frequency of heartbeats. The frequency, height, width, and prominence of the highest peak artifact-free power spectrum ($N_{BVP\perp Acc_t}$) were then calculated and concatenated with the set of all feature maps.

### 2.4. Sample Imputation

When the wristband sensor is not in contact with the skin, biosignals are more likely to be recorded as missing data. Occasionally, data can be missing towards the end of an experiment because subjects may forget to keep their wristbands after the clinical experiment is over.

Truncating biosignals from sample data collected during clinical experiments can negatively impact the richness of sample classes, especially samples recorded during induced mental excitation and psychological stress. Thus, replacing missing samples with meaningful values is necessary to avoid reducing the length of the collected samples. This process is known as imputation. It is possible to impute samples using simple methods, such as mean and median, or to use more advances approaches, such as splines or probabilistic principal component analysis (PPCA) [43–45]. To reconstruct missing values from samples, PPCA with five principal components (PC) was used, then the estimated values were used to replace missing samples.

### 2.5. Feature Selection

Utilizing the ROCKET technique, more than 1800 feature maps for each ten-second slice of biosignals were extracted. One challenge that can be observed in the matrix of feature datasets is the co-linearity of certain features. To resolve this issue, we calculated the Pearson correlation coefficient for pairs of feature variables and dropped features with correlation scores above 0.9. Therefore, those feature variables with the highest co-linearity were excluded from the data. The remaining 1249 feature variables were ranked using three different feature selection techniques. When estimating each target variable, the top 200 features were selected and the pair-wise mutual features were selected for training LSTM NN models, as depicted in the Figure 4.

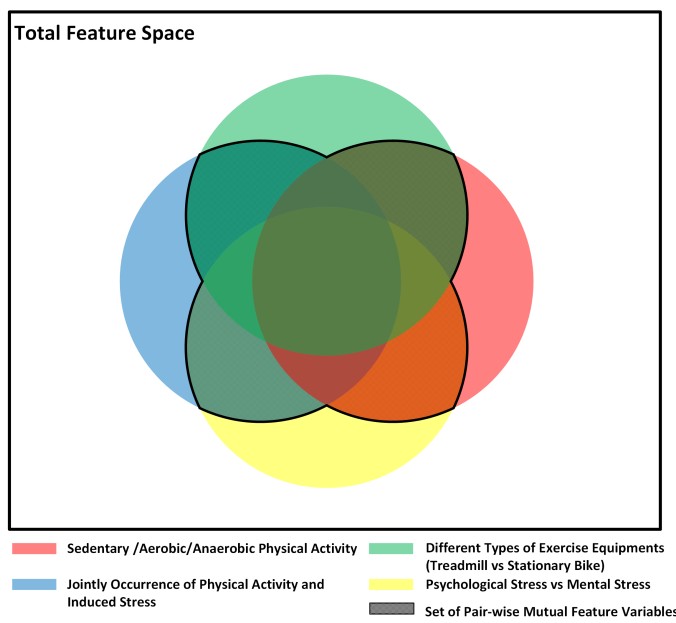

**Figure 4.** Pairwise aggregation of mutual features (the shaded area).

### 2.5.1. Principal Component Analysis

PCA is a widely-used method of analyzing multidimensional data. PCA is an orthogonal projection (or transformation) of the data to a (possibly lower dimensional) subspace [46,47]. PCA reduces the dimensionality of large datasets by creating new un-

correlated latent variables, known as principal components, such that the variance of the projected data is maximized [48,49]. The PC vectors can be found as the eigenvectors of the non-singular portion of the covariance matrix. In this work, we used PCA to reduce the conventional feature space by retaining the features with the highest contributions to a few most dominant PCs. PCA is an unsupervised technique that does not consider the output class labels when selecting the feature variables.

### 2.5.2. Partial Least Squares–Discriminant Analysis

PLS–DA is an algorithm used for predictive modeling and discriminative variable selection. PLS–DA determines the separation among classes and helps to understand which variables contain class-defining information [50,51]. The variable important for projection (VIP) was used in this work to determine the feature variables to retain for building the LSTM model. The VIP summarizes the importance of the variables that explain the regressor feature variables that correlate with the class labels [52]. The VIP was normalized, with a VIP greater than 1 indicating that the feature variable is important for the projection.

### 2.5.3. Sequential Forward Selection

Sequential search algorithms utilize bootstrap-aggregated decision trees as the base classifier and select the most important features in a sequential manner. These algorithms can provide high classification accuracy and have been widely used in various areas. An SFS algorithm starts with an empty set and proceeds by adding one feature variable at a time [53,54]. At each iteration, the SFS algorithm selects the feature variable that is most informative in terms of classification accuracy. By sequentially selecting the feature variables, the SFS algorithm advances one step further towards the optimal classification accuracy. The SFS algorithm adds features one at a time to the set until no further improvement in the results is observed or until a prespecified number of feature variables is reached.

### 2.6. LSTM NN Model

In this work, four RNN classifiers were developed for different tasks:

$Target_{mdl,1} := \{$Non-stress, Emotional Anxiety Stress, Mental Stress$\}$

$Target_{mdl,2} := \{$Sedentary State, Stationary Bike, Treadmill Run$\}$

$Target_{mdl,3} := \{$Sedentary State, Aerobic Exercise, Anaerobic Exercise$\}$

$Target_{mdl,4} := \{$Sedentary State and Non-stress, Sedentary State and Mental/Anxiety stress,

Aerobic/Anaerobic Exercise and Non-stress,

Aerobic/Anaerobic Exercise and Mental/Anxiety stress$\}$

We used NN models with LSTM architecture to perform all classification tasks. LSTMs are a type of recurrent NN (RNN) capable of learning long-term dependencies. An LSTM unit is designed to handle the vanishing gradient and exploding gradients problems, which cannot be solved with ordinary RNNs [55,56]. The LSTM–Dense RNN model used in this study employed several layers, in the following order: an LSTM, a dropout layer, a fully connected layer with dropout, and an output layer with a softmax activation function [57,58].

In the proposed RNN structure, a single LSTM layer with 40 hidden units was used. Parameter tuning with a dropout strategy was applied to the hidden states as well as to the input of the LSTM units. Rectified linear units were used as the activation function of both the LSTM and dense layers. In the RNN structure, a single fully-connected layer with 40 hidden neurons was used. Softmax units were used as an activation function in the output layer for different classification models to predict the probability distribution of target classes (Figure 5). More details on the parameters of the model are provided in Table 3. Figure 5a shows the RNN architecture for the classifications of APS types, device types, PA types, and joint occurrence of PA and APS. Various classification tasks and classes are

listed in Table 4. APS classes included Non-Stress, Emotional Anxiety Stress, and, Mental Stress. AP classes based on equipment type were Sedentary State, Stationary Bicycling, and Treadmill Run. AP types were Sedentary State, Aerobic Exercise and Anaerobic Exercise. The jointly estimated states were centered on APS and PA: Sedentary State and Non-Stress, Sedentary State and Mental Stress or Emotional Anxiety Stress, Aerobic Exercise or Anaerobic Exercise and Non-Stress, and Aerobic Exercise or Anaerobic Exercise and Mental Stress or Emotional Anxiety Stress.

**Table 3.** The value of adjustable parameters used in the NN classifier model.

| Variable | Value/Technique |
|---|---|
| Number of nodes in the dense layer | 40 |
| Number of nodes in the LSTM layer | 40 |
| Number of nodes in the softmax layer | 3 or 4 depending on the classification problem |
| Dropout in the LSTM layer | 20% |
| Dropout in the dense layer | 20% |
| Learning rate | $10^{-5}$ |
| Optimization algorithm | Adam |
| $\beta_1$ | 0.9 |
| $\beta_2$ | 0.999 |
| $\epsilon$ | $10^{-7}$ |
| Size of batches | 10,000 |
| Activation function | ReLU |
| Number of epochs | Variant, depending on the target variable and the size of samples |

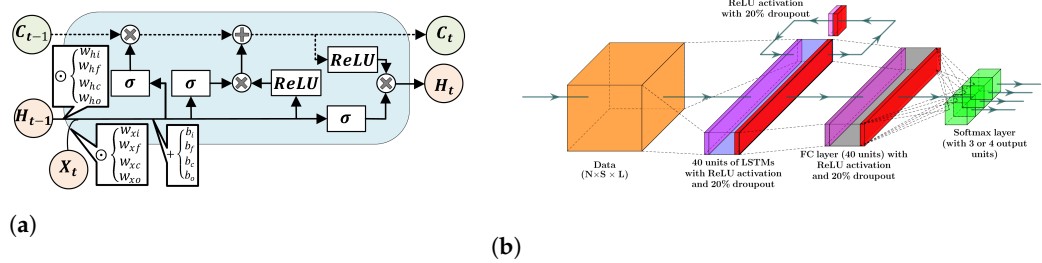

(**a**)

(**b**)

**Figure 5.** (**a**) The structure of a regular LSTM unit and (**b**) the architecture of an LSTM–Dense layer NN model. The size of the softmax layer was set to three units for estimating different variations of APS and PA and four units for estimating both together.

### 2.7. Handling Class Imbalances

Standard LSTM NN models are not capable of handling imbalanced classes, as NNs are trained by calculating errors made by the model on the training splits. Therefore, model weights are updated in proportion to estimation error of training split using the backpropagation algorithm. Thus, samples from different classes are treated equally during training, and the model is trained to classify samples from the majority class, resulting in poor performance for samples from minority classes. To handle the imbalanced classes, we tested four different approaches: upsampling, downsampling, weighted training, and ADASYN [59,60]. Upsampling minority classes is achieved by bootstrapping samples in the minority classes until the number of the training samples in all classes is balanced. In a random downsampling method, majority classes are downsampled by randomly removing samples until the training samples of all classes are balanced. In certain cases, random downsampling can result in the loss of valuable information, resulting in a loss of performance during classification.

In weighted training/cost-sensitive optimization [61,62], the optimizer and the loss function are updated to weigh samples inversely proportional to the number of the samples in each class. Hence, large weights are applied to the samples in the minority class and smaller weights are applied to the samples of the majority class.

The main reason for using ADASYN to balance classes is to generate synthetic samples for minority classes. Therefore, additional synthetic data are generated for minority class samples that are harder to learn compared to those minority samples that are easier to learn. ADASYN is a modified technique of synthetic minority class oversampling (SMOTE) [63,64]; it uses density distribution to adaptively generate a number of synthetic samples, whereas SMOTE generates an equal number of synthetic samples for each minority class. The pseudocode of Algorithm 1 is provided below in order to better summarize the model training pipeline.

---

**Algorithm 1** Pseudocode for training classifier models from raw biosignals

---

1: **procedure** DATA-PREPROCESSING-AND-MODEL-FITTING($BVP$,$Acc_t$,$GSR$,$ST$,$Class_{Labels}$)
2: – Sync all biosignals of Empatica E4 with the corresponding $Class_{Labels}$
3: – For each subject segment $BVP$, $Acc_x$, $Acc_y$, $Acc_z$ $GSR$, $ST$, $Class_{Labels}$ as demonstrated in Figure 2
4:    **for** j = 1:$N_{subjects}$ **do**       ▷ $N_{subjects}$ denote the number of all subjects (datasets)
5:      **for** $k = 1 : N$ **do**       ▷ $N$ denote the number of segments in dataset
6:         – Filter out each segment $Acc_{t,k}^j$, $t \in \{x, y, z\}$ with $4^{th}$ order bandpass filter
7:         – Filter out each segment $BVP^j$ with $4^{th}$ order bandpass filter ([0.2–3.3] Hz)
8:         – Upsample each segment $GSR_k^j$ by two and extract SCL and SCR. (See [65] for more info.)
9:         – Perform ROCKET on each segment of $Acc_{t,k}^j$, $t \in \{x, y, z\}$, $ST_k^j$, $GSR_{tonic,k}^j$, $GSR_{phasic,k}^j$, $BVP_k^j$
10:         – Perform power spectral subtraction by using Equation (3)
11:         – Extract the frequency, height, width, and the prominence from artifact-free segments $N_{BVP \perp Acc_t}$
12:         – Augment feature extracted from the power spectrum $N_{BVP \perp Acc_t}$ to features extracted by ROCKET
13:      **end for**
14:      – Stack each 5 consecutive segmented of features to construct the tensor of input data
15:      – For each subject, stack 3D tensor of features to the rest of tensors of features from other datasets
16:    **end for**
17: – Randomly exclude sessions(labels) different subjects from the stack of the data to split the tensor of input features and target variables into training, validation, and test split
18: – For each one of target variables, perform PCA to select top 200 most informative features
19: – Perform SFS and PLS-DA on each target variable to select top 200 most informative features.
20: – Narrow down the number of selected features from SFS and PLS-DA by keeping mutual features in each pair of target variables
21:
22: Repeat the following procedure for all combination of subset of features, balancing the training split, and the target variable:
23:
24: – Select a subset of feature variables selected from SFS, PLS-DA, or PCA
25: – Select an algorithm from $\{Upsampling, Downsampling, ADASYN, Weighted\text{-}training\}$
26: – Select an array of target classes out of 4 different classification tasks
27: – Balance the training split
28: – Train the LSTM model
29: – Evaluate the model performance
30: **end procedure**

---

## 3. Results

We applied a stratified shuffle split approach to the labels (sessions) for each dataset, with a proportion of 77.25:12.75:10 corresponding to the training, validation, and test sets, respectively. In other words, samples which were collected from the same subjects were only available in training, validation, or split data. The data were randomly selected from sessions performed by each subject, without replacement. The data in the training, validation, and testing splits were taken from different sessions of different subjects, and the

variation of biosignals during different clinical sessions was significantly different. Then, to address imbalanced classes in the training set, we resized each class using the approaches discussed in the previous section. Table 4 summarizes the sizes of the different labels in the training splits of four datasets.

**Table 4.** The size of the training splits after applying different techniques for balancing the training split of the data.

| The size each class in training split for different types of APS inducement | | | | |
|---|---|---|---|---|
| | Upsampling | Downsampling | ADASYN | Initial Size |
| Non-Stress | 101,339 | 21,111 | 100,121 | 23,631 |
| Emotional Anxiety Stress | 101,339 | 21,111 | 101,339 | 101,339 |
| Mental Stress | 101,339 | 21,111 | 100,574 | 21,111 |
| **The size each class in training split for different types of exercise equipment** | | | | |
| | Upsampling | Downsampling | ADASYN | Initial Size |
| Sedentary State | 122,660 | 11,150 | 122,660 | 122,660 |
| Stationary Bike | 122,660 | 11,150 | 123224 | 11,150 |
| Treadmill Run | 122,660 | 11,150 | 122,617 | 11,174 |
| **The size each class in training split for different types of PA** | | | | |
| | Upsampling | Downsampling | ADASYN | Initial Size |
| Sedentary State | 122,661 | 5022 | 12,2661 | 12,2661 |
| Aerobic Exercise | 122,661 | 5022 | 122,489 | 18,407 |
| Anaerobic Exercise | 122,661 | 5022 | 122,596 | 5022 |
| **The size each class in training split for jointly estimation of PA and APS inducement** | | | | |
| | Upsampling | Downsampling | ADASYN | Initial Size |
| Sedentary State& Non-Stress | 111,792 | 10,659 | 111,822 | 10,860 |
| Sedentary State& (Mental Stress or Emotional Anxiety Stress) | 111,792 | 10,659 | 111,792 | 111,792 |
| (Aerobic Exercise or Anaerobic Exercise) & Non-Stress | 111,792 | 10,659 | 111,455 | 11,665 |
| (Aerobic Exercise or Anaerobic Exercise) & (Mental Stress or Emotional Anxiety Stress) | 111,792 | 10,659 | 112,059 | 10,659 |

In order to better compare the performance of the RNN models in predicting class labels of different classification problems, we used the balanced accuracy score (Equation (4)):

$$
\begin{aligned}
bal\,Acc &= \frac{\sum_{i=1}^{n} Specificity_i + Sensitivity_i}{2n} \\
Sensitivity &= \frac{\sum TP}{\sum TP + \sum FN} \\
Specificity &= \frac{\sum TN}{\sum TN + \sum FP}
\end{aligned}
\tag{4}
$$

Balanced accuracy score is the proper metric for coping with fallacious interpretation of imbalance classes in the data [66–68]. The balanced accuracy score and number of misclassified samples are reported in Tables 5–8.

All numerical studies were conducted in the TensorFlow 2.0 environment, and several other Python libraries were used for data preprocessing [40,65,69–71].

**Table 5.** Balanced accuracy score and the number of misclassified samples (values in parentheses) for joint prediction of PA and APS.

|         | Upsampling    | Downsampling   | Weight Training   | ADASYN        |
|---------|---------------|----------------|-------------------|---------------|
| PCA     | 99.49% (68)   | 98.59% (556)   | 99.52% (68)       | 99.40% (83)   |
| PLS-LDA | 99.68% (39)   | 99.11% (383)   | **99.80% (35)**   | 99.64% (47)   |
| SFS     | 98.59% (294)  | 96.82% (1145)  | 96.95% (845)      | 97.43% (249)  |

**Table 6.** Balanced accuracy score and the number of misclassified samples (values in parentheses) for predicting different types of PA (Aerobic Exercise(AE)/Anaerobic Exercise(ANE)/Sedentary State).

|         | Upsampling    | Downsampling   | Weight Training   | ADASYN        |
|---------|---------------|----------------|-------------------|---------------|
| PCA     | 99.8% (20)    | 98.84% (364)   | 99.89% (32)       | 99.88% (20)   |
| PLS-LDA | **99.99% (2)**| 99.65% (144)   | 99.97% (5)        | 99.99% (4)    |
| SFS     | 99.28% (80)   | 97.37% (666)   | 99.16% (203)      | 98.82% (88)   |

**Table 7.** Balanced accuracy score and the number of misclassified samples (values in parentheses) for predicting different types of APS (Non-Stress/Mental Stress/Emotional Anxiety Stress).

|         | Upsampling    | Downsampling   | Weight Training   | ADASYN        |
|---------|---------------|----------------|-------------------|---------------|
| PCA     | 99.32% (119)  | 98.10% (543)   | 99.28% (123)      | 98.88% (216)  |
| PLS-LDA | 99.56% (68)   | 99.04% (273)   | 99.59% (66)       | **99.6% (85)**|
| SFS     | 98.15% (312)  | 97.35% (617)   | 97.86% (504)      | 97.69% (326)  |

**Table 8.** Balanced accuracy score and the number of misclassified samples (values in parentheses) for predicting different types of equipment (Sedentary State/Treadmill Run/Stationary Bike).

|         | Upsampling    | Downsampling   | Weight Training   | ADASYN        |
|---------|---------------|----------------|-------------------|---------------|
| PCA     | 99.87% (11)   | 99.66% (135)   | 99.94% (10)       | 99.92% (10)   |
| PLS-LDA | 99.82% (4)    | 99.82% (78)    | 99.93% (5)        | **99.93% (3)**|
| SFS     | 99.59% (49)   | 98.96% (301)   | 99.49% (126)      | 99.38% (59)   |

## 4. Discussion of Results

This work addresses the related problems of classifying possible occurrences of various types of physical activity, acute psychological stress, and their simultaneous occurrence. The measurement of hormones or biochemical markers during various types and intensities of PA and APS is not feasible in free-living conditions. This limits the detection of PA and APS to the few measurements that are readily measurable using noninvasive sensors, typically in a form factor acceptable for daily long-duration wear and use, such as a wristband. This is a challenging problem because the functional responses to PA and APS can affect the same noninvasively measured physiological variables, such as changes in the heart rate and heart rate variability. Thus, detecting the simultaneous occurrence of PA and APS using only a few noninvasively measured physiological variables represents a challenge for conventional techniques.

This work uses multiple physiological variables and captures the information by treating these signals with feature extraction, feature variable selection, and techniques for handling imbalanced classes to ensure that accurate predictions of the type of PA and APS are rendered for accurate classification of events and optimal insulin therapy decisions in people with diabetes. We integrate several physiological variables reported by the wristband with ML algorithms to classify the types and intensities of individual or simultaneously-occurring PA and APS. Recognizing that the raw data reported by the physiological variable sensors are susceptible to measurement noise, missing data, corrupting artifacts, and other inconsistencies, we first preprocess the data to improve the signal quality. Following preprocessing, we generate feature variables from the signal, and select the most informative feature variables for the classification tasks. We evaluate

three different feature selection methods, namely, a PCA-based approach, a technique utilizing the VIP metric in PLS-DA, and SFS.

For each feature selection technique, we evaluate four different approaches to handle imbalances in the sizes of the different classes, which can bias the machine learning algorithms. Because data from sedentary states (resting) and non-stress states are much easier to collect, these classes may be overrepresented in the data. For each feature selection approach, we evaluate four different algorithms for handling imbalanced classes: down-sampling the overrepresented classes by removing samples, up-sampling the overrepresented classes by sampling with replacement, weighting the errors with weights inversely proportional to the respective class size, and ADASYN sampling.

Recognizing the temporal dependencies in the data, we leverage the LSTM architecture, a type of RNN that is capable of learning order dependence in a sequence of prediction. This systemic study elucidates the abilities of various algorithms for feature selection and the different techniques for handling imbalanced class sizes.

The results reported in Tables 5–8 indicate that the balanced accuracy scores for various combinations of data balancing and feature selection techniques range between 96.82% and 99.99%. The number of misclassified samples has a larger variation based on the techniques used, with PLS-DA having the smallest number of misclassifications. The main advantage of PLS-DA over the PCA-based approach is that PLS-DA is a supervised method that encodes the relationships between the large number of extracted features and the target class affiliations composed of the true class labels. Sequential feature selection approaches may perform better when incorporating floating techniques that allow backtracking for an arbitrary number of times, as this facilitates the discarding of one feature at a time for a certain number of total discarded features for as long the quality criterion continues to improve. The use of sequential floating search methods will be further explored in future works. A limitation of sequential feature selection techniques is the absence of selection using orthogonal subspaces, which permits collinearity in features selection. The imbalances in class sizes were well handled by the ADASYN sampling approach, which has the desired properties of reducing bias caused by class imbalances and adaptively transitioning the decision boundary towards the more challenging classification examples.

The participants in our study included healthy subjects and subjects with Type 1 diabetes. Acute psychological stress inducements that are accepted by the Institutional Review Boards were limited both in type and intensity. To obtain a large enough dataset, in this study we use data from both healthy and diabetes groups. The developed algorithm is based on information reporting body movement (3D accelerometer signal), body temperature, skin conductivity, and cardio vascular activity (BVP signal). These variables are signals directly recorded by sensors and do not depend on the glycemic condition of subjects. Hence, they indicate the state of the body, namely, activation of the sympathetic and parasympathetic nervous systems (i.e., the presence of acute psychological stress), physical activity, and other states of the body that are not covered in this manuscript, such as sleep characteristics. Therefore, the acute psychological stress detection model presented here is quite general, as it has been tested with samples from both healthy and diabetes subjects. Of course, the effect of acute psychological stress on blood glucose levels is different for subjects without and with diabetes. Glucose concentration estimation for people with Type 1 diabetes is addressed in [72].

This work can benefit people with diabetes relying on insulin administration as a therapy to manage their blood glucose concentrations. People with T1D must make frequent adjustments to their insulin therapy throughout the day based on the various physical and psychological states they experience. Although accommodating structured planned exercise is relatively straightforward, as advance information on the time and intensity of the planned future exercise can be used to manually adjust insulin infusion policies, spontaneous unplanned physical activities that occur throughout the day are challenging for people to accommodate in their insulin dosing decisions. Moreover, episodes of acute psychological stress can occur concurrently or independently of PA, such as during stressful

driving conditions or running to catch a bus to work, and may be difficult to explicitly accommodate through insulin therapy adjustments. This problem is further complicated by the fact that PA typically reduces BGC (increasing the risk of hypoglycemia), while APS can cause a temporary increase in BGC (increasing the risk of hyperglycemia). Confounding the occurrences of PA with APS may lead to incorrect insulin therapy decisions that exacerbate the glycemic excursion. Using only a limited number of physiological measurements can therefore lead to incorrect assessments of a person's physical and psychological state, and consequently to inappropriate insulin therapy decisions. Applying machine learning to the physiological variables collected from a wearable device generates accurate detection and classification of PA and APS information. Our future work will integrate PA and APS information with artificial pancreas systems for provision of precision medicine through optimal automated insulin delivery in people with diabetes [2,3].

## 5. Conclusions

In this work, we have addressed problems involved in detecting the occurrence of individual and concurrent PA and APS events, as well as in differentiating types of PA and APS from data collected with an Empatica E4 wristband. We employed convolutional-based feature extraction by performing random kernel filters on each biosignal to elicit pattern in the signals during different kinds of PA and APS. We evaluated different feature selection algorithms to select a subset of features for discriminating between different types of PA and APS. We then tested different resampling techniques to mitigate the issue of imbalanced classes. The balanced accuracy scores indicate that the ROCKET feature extraction algorithm and suggested sample stacking algorithm are excellent choices for time series classification of PA and APS. The combination of PLS–LDA for feature selection and ADASYN for data balancing provides the best overall performance among all of the considered detection and classification cases. The use of such techniques can provide reliable information on both daily behavior patterns and unexpected events, thereby assisting in treatment that is effective in mitigating the effects of events that can disturb metabolic homeostasis.

**Author Contributions:** Conceptualization, M.R.A., M.R. and A.C.; methodology, M.R.A., M.R. and M.S.; software, M.R.A. and M.A.-L.; validation, M.R.A. and M.A.-L.; formal analysis, M.R.A., M.R. and A.C.; investigation, M.R.A., M.S., M.A.-L.; resources, M.R.A., M.R. and A.C.; data curation, M.S. and M.R.A.; writing—original draft preparation, M.R.A., M.R. and A.C.; writing—review and editing, M.R.A., M.R. and A.C.; visualization, M.R.A. and M.A.-L.; supervision, M.R. and A.C.; project administration, A.C.; funding acquisition, A.C. All authors have read and agreed to the published version of the manuscript.

**Funding:** Financial support from the NIH under the grants 1DP3DK101075 and 1R01DK130049 and the JDRF under grant 2-SRA-2017-506-M-B (made possible through collaboration between the JDRF and The Leona M. and Harry B. Helmsley Charitable Trust) is gratefully acknowledged.

**Institutional Review Board Statement:** This study was conducted in accordance with the Declaration of Helsinki and approved by the Institutional Review Board of Illinois Institute of Technology (Protocol code IRB 2019-018, date 16 October 2019).

**Informed Consent Statement:** Informed consent was obtained from all subjects involved in the study.

**Data Availability Statement:** The models were developed in open-source Python libraries and the trained models are available at the GitHub public repository https://github.com/rezaaskary/MDPI-Algorithms.git (accessed on 31 July 2022).

**Acknowledgments:** Ali Cinar is grateful for funds provided by the Hyosung S. R. Cho at the Illinois Institute of Technology. Research reported in this publication was partially supported by the National Institute of Diabetes And Digestive And Kidney Diseases of the National Institutes of Health under Awards No. 1DP3DK101075 and 1R01DK130049. The content is solely the responsibility of the authors and does not necessarily represent the official views of the National Institutes of Health.

**Conflicts of Interest:** The authors declare no conflict of interest. The funders had no role in the design of the study, in the collection, analysis, or interpretation of the data, in the writing of the manuscript, or in the decision to publish the results.

## Abbreviations

The following abbreviations are used in this manuscript:

| | |
|---|---|
| APS | Acute Psychological Stress |
| PA | Physical Activity |
| ROCKET | Random Convolutional Kernel Transformation |
| BVP | Blood Volume Pulse |
| GSR | Galvanic Skin Response |
| ST | Skin Temperature |
| PCA | Principal Component Analysis |
| SFS | Sequential Forward Selection |
| PLS-DA | Partial Least Squares–Discriminant Analysis |
| ADASYN | Adaptive Synthetic Sampling |
| LSTM | Long Short-Term Memory |
| BGC | Blood Glucose Concentration |
| T1D | Type-1 Diabetes |
| CGM | Continuous Glucose Monitoring |
| HR | Heart Rate |
| IBI | Inter-beat Interval |
| SCR | Phasic Skin Conductance Response |
| SCL | Tonic Skin Conductance Level |
| PPV | Proportion of Positive Values |
| FFT | Fast Fourier Transform |
| PPCA | Probabilistic Principal Component Analysis |
| PC | Principal Components |
| VIP | Variable Important for Projection |
| ReLU | Rectified Linear Units |
| SMOTE | Synthetic Minority Oversampling Technique |

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
