# Peer review of "Detection and Classification of Unannounced Physical Activities and Acute Psychological Stress Events for Interventions in Diabetes Treatment"

_algorithms, doi:10.3390/a15100352_

Round 1
Reviewer 1 Report
This paper proposes a method to detect psychological stress and physical activity using biosignals extracted from a wristband. Authors explore multiple feature selection and resampling techniques to mitigate the issue of class imbalance, and propose the use of a LSTM model to perform the classification task. The paper introduces several potential contributions to knowledge, however several improvements and/or clarifications are required, according to the comments below:
1- The Abstract misses a clear presentation of the motivation and context of the study. I would recommend motivating the reader for the work already in the Abstract.
2- The paper is not clear about which classes were used in the classification task. The Abstract, for instance, does not provide any information about it. We understand from Figure 5 that you formulated the problem as a multi-class classification, however, it is not clear what should be the output in each of the (3 or 4?) output units. The methods section does not state which classes were considered to formulate the problem. Moreover, it is not clear if you trained your classifier (multiple times) for different problems (e.g., to classify different activities, or to classify different stress levels, etc.), or if you predicted activities and stress levels jointly. Please clarify.
3- In the Introduction (lines 62-67) you mention some previous works. What has been the performance achieved in these previous works? How does this paper differentiates from past works in this field?
4- On the Methods section, you miss to describe what was the sampling rate of the biosignals extracted from Empatica E4.
5- In Figure 1, you describe the signals used to discriminate PA and APS as being in red and italics. However, italics has not been used.
6- In lines 105-109 you state that you did not use the variables IBI and HR, and justified it by the fact that PPG is affected by noise due to intense body movements. But what about BVP, isn't it estimated from PPG as well? If these values are not reliable during intense physical activity, why not using them when the levels of activity are low?
7- Please provide references for the normal values of respiration rate and HR stated in lines 125 to 127. Also for the types of electrodermal activity that you mention in lines 137 to 144.
8- In line 129 you mention a bandpass filter. What filter did you use?
9- You have applied different filtering cutoffs to different signals. But, what about the filtering delays? Have you compensated for the delays introduced by the filters? Considering that different filters may add different delays to the signals, were signals still synchronized after filtering? Also, were they synchronized before filtering?
10- In lines 144 to 146 you mention that "The preprocessing of GSR consists of upsampling the signal and estimating the baseline of the signal and differentiating it from the signal to extract SCL and SCR, respectively." Why did you need to perform the upsampling and to what sampling frequency? What do you mean by "differentiating it from the signal"?
11- In lines 152-153, you mention that "integrating convolutional layers with LSTM architecture cannot be done efficiently with GPUs". However, the literature has many examples of combining convolutional and LSTM layers. Could you please clarify on the motivation to use ROCKET?
12- What were the inputs of your LSTM? I understood that you extracted several features from each 10 second-window. What was the dimension of your input data? Have you taken advantage of the recurrent / sequential nature of LSTMs? Were your inputs still time-series, or just a bunch of features representing each 10s window? And what about the output, did you predict the class of each window? Have you applied any post-processing to ensure some continuity and consistency between predicted classes in adjacent windows?
13- In section 2.3 you mention the sample imputation. Did this happen after feature extraction? Wasn't it applied to the raw sensor data (i.e., applied as part of the pre-processing stage)? If not, how did you handle missing data while extracting features?
14- In Figure 3 you should identify which steps correspond to the pre-processing step. Moreover, since you did not use the signals represented in red, I would suggest removing them from the figure.
15- In line 190 you mention that "feature maps with the highest co-linearity were excluded from the data". How did you evaluate co-linearity?
16- Could you please clarify the statement in lines 191-193? ("For estimating each target variable, the top 200 features were selected and the pair-wise mutual features were selected for training LSTM NN models as depicted in the figure 4").
17- For the Sequential Forward Selection, what LSTM model have you used? The same model used for the classification task? What about the sentence in lines 221-223 ("The SFS algorithm adds features one at a time to the set until no improvement in the results is observed anymore or until a prespecified number of feature variables is reached"), have you extracted a predefined number of features, or did you stop when no more improvements were observed?
18- In the Results section you start by describing the dataset. I would recommend moving the dataset collection to the previous section, as data collection is still part of the Methods, not the Results.
19- The dataset description is very poor, and should be improved. The characteristics of the participants (e.g., age, sex, diseases, etc.) are not mentioned, and should be thoroughly analyzed as these characteristics may have an impact on the measured biosignals.
20- Throughout the paper, you use a lot of acronyms. I would suggesting maintaining only the ones most commonly used on the state of the art, because having too many acronyms slows down the reading and hinders the comprehension of the paper.
21- In Table 2 you present the characteristics of the training split. What about the testing split, how many samples were there? Also, have you included data from the same subjects on the training and test sets? How can it impact your results (e.g., concerning overfitting)? What about the validation split? For what purpose have you used it?
22- Why did you use a balanced accuracy score? Since you applied methods to balance the representation of your classes in the dataset, shouldn't balanced accuracy be equivalent to accuracy?
23- You paper does not mention which parameters were used on the training of the NN, for instance, number of epochs, batch size, optimization algorithm, etc. Could you please clarify?
24- The Discussion of your work misses very important points, for instance, concerning the results achieved (e.g. what was the best performing algorithm, which classification task had the best results, what are the advantages of estimating PA jointly with APS, etc.), comparison with other studies (concerning methods and results; advantages and disadvantages; etc.), limitations of your work, contributions, future work, etc. In the Discussion, you should thoroughly analyze and interpret your findings.
25- Please consider reviewing the references you used to support your work. A total of 79 references are included, but certainly not all of them are needed to support you work. Please carefully review it and just stick to the most relevant ones.
Author Response
The review was very helpful in enhancing the manuscript. Thank you for the time and effort you invested in reviewing the manuscript.
Please see the attachment that provides point by point replies to your comments.

Reviewer 2 Report
The manuscript is quite well written. However I suggest to better show how the proposed technique advances the state of the art, and compare it to other similar techniques.
Additionally, a flow chart of the overall methodology must be displayed.
Finally, a pseudocode of the procedure must be provided.
Author Response
Thank you for your comments and suggestions to improve the manuscript. The changes made are shown by comparing the PDF files of the initial and the revised manuscript and attached as a supplementary file (not for publication). The changes made and response to specific comments are listed below.
The manuscript is quite well written. However, I suggest to better show how the proposed technique advances the state of the art, and compare it to other similar techniques.
The introduction section is modified to address the comment of the reviewer.
Additionally, a flow chart of the overall methodology must be displayed.
Figure 3 and Figure 5 were used to display the pipeline of the methodology.
pseudocode of the procedure must be provided.
The pseudocode has been added (Alorithm1)
Round 2
Reviewer 1 Report
Thank you for addressing some of my concerns and providing a point-by-point response to my comments.
Although some of the points are addressed in your response, I would like some of the provided clarifications to be included in the text. In particular:
- which bandpass filter you used to filter BVP (as questioned in my previous point 8);
- a more detailed explanation about the GSR signal preprocessing (as questioned in my previous point 10)
- how co-linearity was evaluated (questioned in my previous point 15)
- which classifier was used in the sequential forward selection algorithm (questioned in my previous point 17)
Additionally, the following concerns remain open:
1. Concerning the participants in your study, were they healthy, or did they suffer from diabetes? In case they were healthy, how do you expect the algorithm to behave in patients with diabetes? Will it generalize well?
2. In lines 286-288, please clarify how many classification tasks and what classes have you considered in each classification task ("Figure 5a shows the RNN architecture for the classifications of APS types, device types, PA types, and joint occurrence of PA and APS.")
3. In my previous question 21, you mentioned that a "random number of samples from each subject is included in all three splits". By knowing this, I would conclude that your good results are, in fact, just a consequence of overfitting due to the fact that the algorithm is being tested on the same subjects included in training. This can, thus, be a major flaw of your approach. Could you please comment or justify this?
Reviewer 2 Report
The authors did a good job addressing the reviewers' suggestions and comments. Therefore, now I can recommend to accept this paper as it is.
Author Response
Thank you for your review and comments. They have been useful in improving the manuscript.
Round 3
Reviewer 1 Report
Thank you for addressing my comments. Only one point remains open. It concerns the characteristics of the participants. As you stated in your answers to my previous comments, you included a combination of healthy and diabetic patients. This information is currently not available in the paper. Please clearly state on the text how many healthy and type 1 diabetes patients you have included in your experiments. Additionally, the discussion you added to my previous question 1 (concerning the generalization to the results) could be added to the Discussion.
I have no additional concerns.
Author Response
Thank you for the suggestions to provide details about subjects and the effects of their status on stress detection.
Information about subjects added:
line 120-121: Six were healthy subjects and 28 were people with T1D.
Updated paragraph in Discussion of results:
lines 390-404:
The participants in our study included healthy subjects and subjects with Type 1 diabetes. Acute psychological stress inducements that are accepted by the Institutional Review Boards are limited both in type and intensity. To have a large enough dataset, we have used the data from both healthy and diabetes groups. The algorithm is developed based on the body movement (3D accelerometer signal), body temperature, skin conductivity of the body, and cardio vascular activity of the body (BVP signal). These variables are signals directly recorded by sensors and regardless of the glycemic medical condition of subjects. Hence, they indicate the state of the body namely, activation of sympathetic and parasympathetic nervous system (presence of acute psychological stress), physical activity, and other states of the body that are not covered in this manuscript (such as sleep characteristics). Therefore, the acute psychological stress detection model is quite general as it has been tested with samples from both healthy and diabetes subjects. Of course, the effect of acute psychological stress on blood glucose levels will be different for subjects without and with diabetes. Glucose concentration estimations for people with Type 1 diabetes is addressed in [73].